# Vagus Nerve Stimulation: A Personalized Therapeutic Approach for Crohn’s and Other Inflammatory Bowel Diseases

**DOI:** 10.3390/cells11244103

**Published:** 2022-12-17

**Authors:** Giovanni Cirillo, Flor Negrete-Diaz, Daniela Yucuma, Assunta Virtuoso, Sohaib Ali Korai, Ciro De Luca, Eugenijus Kaniusas, Michele Papa, Fivos Panetsos

**Affiliations:** 1Division of Human Anatomy, Neuronal Morphology Networks & Systems Biology Lab, Department of Mental and Physical Health and Preventive Medicine, University of Campania “Luigi Vanvitelli, 80138 Naples, Italy; 2Neurocomputing & Neurorobotics Research Group, Universidad Complutense de Madrid, 28040 Madrid, Spain; 3Instituto de Investigaciones Sanitarias (IdISSC), Hospital Clinico San Carlos de Madrid, 28040 Madrid, Spain; 4Andalusian School of Public Health, University of Granada, 18011 Granada, Spain; 5Institute of Biomedical Electronics, TU Wien, 1040 Vienna, Austria; 6SYSBIO Centre of Systems Biology ISBE-IT, University of Milano-Bicocca, 20126 Milan, Italy; 7Silk Biomed SL, 28260 Madrid, Spain

**Keywords:** inflammatory bowel disease, Crohn’s disease, vagus nerve, vagus nerve stimulation, neuroprostheses, auricular vagus nerve, transcutaneous auricular vagus nerve stimulation

## Abstract

Inflammatory bowel diseases, including Crohn’s disease and ulcerative colitis, are incurable autoimmune diseases characterized by chronic inflammation of the gastrointestinal tract. There is increasing evidence that inappropriate interaction between the enteric nervous system and central nervous system and/or low activity of the vagus nerve, which connects the enteric and central nervous systems, could play a crucial role in their pathogenesis. Therefore, it has been suggested that appropriate neuroprosthetic stimulation of the vagus nerve could lead to the modulation of the inflammation of the gastrointestinal tract and consequent long-term control of these autoimmune diseases. In the present paper, we provide a comprehensive overview of (1) the cellular and molecular bases of the immune system, (2) the way central and enteric nervous systems interact and contribute to the immune responses, (3) the pathogenesis of the inflammatory bowel disease, and (4) the therapeutic use of vagus nerve stimulation, and in particular, the transcutaneous stimulation of the auricular branch of the vagus nerve. Then, we expose the working hypotheses for the modulation of the molecular processes that are responsible for intestinal inflammation in autoimmune diseases and the way we could develop personalized neuroprosthetic therapeutic devices and procedures in favor of the patients.

## 1. Introduction

Autoimmune diseases occur when the immune system (ImS) does not discriminate correctly between molecules that are self and molecules that are non-self to the organism. When this happens, ImS attacks its own cells as if they were non-self. Most of these diseases are accompanied by chronic inflammation, such as systemic lupus erythematous, type I diabetes mellitus, or inflammatory bowel disease (IBD) [1]. IBD is a generic term that pools several autoimmune diseases, including Crohn’s disease and ulcerative colitis, both of them characterized by chronic inflammation that affects the gastrointestinal tract [2]. The etiopathogenesis is unknown, but there is increasing evidence that it is multifactorial and associated with genetic and environmental risk factors. 

A key role in IBD pathogenesis is deserved to the central nervous system (CNS) and its interactions with the enteric nervous system (ENS). ENS is embedded within the gut wall and is widely interconnected with the CNS, the enteroendocrine and gastrointestinal ImS, the gut microbiota, and the peripheral nervous systems (PNS). ENS regulates gastrointestinal motility, nutrient absorption, secretion, immunity, and defense [3]. A failure in ENS-CNS communications may be involved in ImS deregulation and provoke (or contribute to) IBD pathogenesis [4]. There is increasing evidence that IBD is correlated with a low tone of the vagus nerve (VN), a parasympathetic branch of the CNS strongly involved in CNS-ENS communication [5].

Therefore, the restoration of CNS-ENS-ImS impairments by modulating the output of the VN could lead to a therapeutic approach for IBD [6,7,8]. This hypothesis is also supported by recent experimental results showing that external modulation of VN efferent signals is efficient in reducing inflammation in diverse conditions, spanning from rheumatoid arthritis to sepsis and IBD [9,10].

In the present paper, we provide a comprehensive overview of the cellular and molecular bases of CNS-ENS interactions and the way these interactions could affect the ImS that underlies IBD. Furthermore, we discuss the therapeutic application of electrical stimulation of the vagus nerve (VNS), with an emphasis on the electrical stimulation of the auricular branch of the vagus nerve, highlighting the action mechanisms that modulate the molecular processes responsible for intestinal inflammation in IBD patients [11,12,13].

## 2. The Immune System

ImS protects the organism against harmful and potentially detrimental agents that could trigger disease onset and progression through two different defense strategies: the innate and the acquired response (Figure 1) [14]. 

These responses are cell-mediated (e.g., macrophages, lymphocytes, natural killers) or molecule-mediated (e.g., cytokines, complement factors, antibodies) [15]. Foreign or self-molecules, called antigens, are detected by specific receptors localized on the surface of the immune cells and activate the ImS [16], triggering a cascade of events that boosts an inflammatory response and culminates with the neutralization of the antigen [17]. ImS activation also leads to the release of defense molecules called antibodies or immunoglobulins that identify and neutralize antigens through the high-affinity binding of a specific part of the antigen (called epitope) [14] and generate an “immunological memory” against antigens [18].

The innate immune response represents the first line of defense of the ImS. It is an antigen-independent defense mechanism that acts non-specifically and immediately (or within several hours) to eliminate microbes and prevent infection [19]. Innate immunity can be divided into immediate innate immunity and early induced innate immunity. The first begins 0–4 h after exposure to an infectious agent and involves the action of soluble preformed antimicrobial circulating molecules or present in extracellular fluids or secreted by epithelial cells (antimicrobial enzymes and peptides, complement system proteins) but also anatomical barriers to infection, mechanical removal of microbes, and bacterial antagonism by normal body microbiota. Early induced innate immunity begins 4–96 h after exposure to an infectious agent and is based on the release of inflammatory mediators and recruitment of defense phagocytic cells such as leukocytes (neutrophils, eosinophils, and monocytes), tissue phagocytic cells (dendritic cells, macrophages, and mast cells), and natural killer cells [20,21]. Unlike adaptive immunity, innate immunity does not recognize every possible antigen but key molecules of the pathogens, such as lipopolysaccharide (LPS) from the gram-negative bacterial wall or peptidoglycan from the gram-positive cell wall, the sugar mannose (common in microbial glycolipids and glycoproteins but rare in human ones), bacterial and viral unmethylated CpG DNA, bacterial amino acid or protein such as N-formylmethionine or flagellin, single/double-stranded viral RNA, and glucans from fungal cell walls.

The acquired immune response is mediated by cells (lymphocytes T or B) and antibodies. T cells express surface receptors that allow them to detect and generate responses to various agents whilst B lymphocytes are responsible for the antibody-mediated responses. Antigen-presenting cells (APCs) allow B cells to produce specific antibodies against the identified antigen; in addition, like APCs, B cells can also present the antigen to T cells initiating an adaptive immune response [22]. In addition to B cells, macrophages and dendritic cells can act as APCs and integrate cues from epithelial, immune, stromal, and neural cells to direct innate and adaptive immunity [22,23,24]. Once antigen molecules are identified, sensory nociceptive neurons can produce calcium influx and action potentials [15] that can travel toward the spinal cord and brain (orthodromic) or travel toward the periphery (antidromic) as an axonal reflex. These axon reflexes lead to the local release of neuromodulators, including substance P (sub-P), calcitonin gene-related peptide (CGRP), and other molecules that modulate inflammation, impair neutrophil function, and suppress responses to bacteria [15,25]. The nervous system plays a pivotal role in the regulation and modulation of inflammation and immune response, especially through the parasympathetic system [15,26]. 

Inflammation is a consequence of ImS activation and represents a defensive reaction: after antigen identification, it attracts defense cells of the ImS and grants access to the site of damage [16]. However, this process is self-limited and finely regulated to avoid chronicity, a condition that impedes physiologic repair and could be the starting point of an autoimmune disease. Innate and acquired responses generally work together to eliminate antigens even though they are governed by different mechanisms [19,27].

## 3. Interaction between the Immune System and the Enteric Nervous System

The bowel is innerved by an intrinsic and an extrinsic system, provided by ENS and CNS, respectively [28,29]. ENS is organized into two plexuses, the myenteric (Auerbach plexus) and the submucosal ones (Meissner plexus) [29]), and modulates intestinal inflammation and gut motility, among others. These functions are influenced by CNS through the sympathetic and parasympathetic nerves [30] that allow gut information to reach the brain and the spinal cord through neuronal afferent pathways [31,32,33] (Figure 2).

Below the intestinal mucosa, a network of neurons and glia, called the enteric nervous system (ENS), is responsible for the regulation of all sensory and motor functions within the gastrointestinal tract and independently of the brain [34,35] (Figure 3).

Enteric neurons synthesize and release a wide range of neurotransmitters and neuropeptides, allowing them to communicate with smooth muscles, epithelium, other neurons, and immune cells [36]. Importantly, where neurotransmitters are stored and released is critical to their function.

ENS regulates intestinal motility [3,37] through interneurons, intrinsic primary afferent neurons (IPANs), and choline acetyltransferase (ChAT)-producing excitatory and nitric oxide (NO) synthase (NOS)-producing inhibitory motor neurons [38]. Within intestinal mucosa, IPANs connect with interneurons which synapse with motor neurons and play a key role in the immune modulation of intestinal motility [6]. To achieve modulation of intestinal inflammation, ENS forms “close associations” and functional interactions with immune cells [36,39,40,41]. Enteric neurons directly secrete immune mediators (i.e., IL-8), express TNF-α and IL-1β receptors, and their activities are altered by pro-inflammatory cytokines (TNF-α, IL-1β, IL-6, and other neuromediators) released by immune cells during intestinal inflammation. IL-1β, for example, can depolarize the membrane, decrease membrane conductance, and increase the discharge of action potentials [42], suggesting that neuronal activity could be influenced by inflammation [43,44] and could modulate or increase the severity of inflammatory responses [36,45].

While enteric neurons lie below intestinal crypts within the submucosa as well as between longitudinal and circular smooth muscle layers, enteric glial cells are located within the intestinal layers, around crypts, and throughout intestinal villi and are involved in the synaptic transmission into the myenteric plexus [29], in the regulation of the intestinal epithelial barrier through cytokine production and regulation of the expression of tight junction-associated proteins [35]. Evidence suggests that enteric glial cells are active during intestinal inflammatory and immune responses, acting as APCs, and they modulate the mucosal ImS via the expression of cytokines and cytokine receptors [46,47,48,49,50,51,52,53]. These cells communicate with each other through changes in intracellular Ca^+2^ levels by communicating junctions and with neurons, especially by purines [54]. Projections from both enteric neurons and glia come into direct contact with the epithelium through junctions with enteroendocrine sensory cells (EECs) residing in the epithelium. This requires cellular proximity, and although conventionally believed to be paracrine communication, direct synaptic connections have been observed through neuropods. EECs respond to chemical and possibly mechanical stimuli within the lumen and secrete hormones and neurotransmitters in response.

The most abundant subclass are enterochromaffin (EC) cells that release 5-HT with a controversial effect between pro and anti-inflammatory, being in the case of enteric neurons an anti-inflammatory neurotransmitter that exerts neuroprotective and neurogenic effects for the CNE to survive inflammation [55].

Evidence suggests extensive bidirectional crosstalk between enteric neurons and enteric glial cells and between the epithelium and ENS, playing a role in the maintenance of intestinal epithelial barrier integrity [46] and control of inflammatory processes [29,36], suggesting that enteric glial cells might also be linked to the ImS. 

Therefore, these interactions may also excerpt an effect on inflammatory processes within the intestine. This close interaction between ENS cells and ImS has led to a novel concept of neuroimmune cell units at the core of gut homeostasis and defense [33]. However, the role of neuroimmune cell units in gut physiology and its implications in gastro-intestinal disease pathogenesis raise open questions, and further evidence needs to be provided.

## 4. Gut Microbiota-Central Nervous System-Immune System Interaction

The gut represents a perfect environment for the life of a very large number of microorganisms (around 10^14^), which not only reside in the interior of the digestive tube but also interact with the host organism, participating in important functions like nutrition, immune responses, brain activation, and contributing to the homeostasis between systems [56,57]. Gut microbiota, whose composition is determined by several factors like genetics, diet, residence location, age, and gender, plays an important role in the maintenance of intestine health, participating in the synthesis of vitamins and the catabolism of indigestible carbohydrates and fatty acid chains, producing short-chain fatty acids (SCFAs), which promote gut barrier integrity, mucus production, and fostering a tolerogenic response to inflammation. On the other side, competitive colonization of the commensal microbiota prevents gut colonization by pathogens [57].

Microbiota modulates ImS development, and ImS, in turn, determines the composition and function of microbiota populations. Alterations of the gut microbiome can lead the host organism to a pathological state [58]. Indeed, germ-free animals are characterized by the underdevelopment of gut-associated lymphoid tissues, reduced immunoglobulin secretion, and reduced intraepithelial CD8+ T cells; interestingly, these animals also present weak peripheral ImS with reduced peripheral CD4+ T cells, including Th17 cells and T regulatory (Treg) cell compartments [57,58].

CNS, ImS, and gut microbiome communicate among them through several pathways [56,57,59]: ImS employs pattern recognition receptors (PRR) to recognize molecular patterns (MAMPs) on the surface of specific microbiota, and the expression of these receptors is modulated by the presence of specific pathogens [60]. Germ-free mice display a low number of enteric neurons, correlated with low gut motility and low neural excitability, suggesting microbiota could modulate the development of the ENS. Short-chain fatty acids affect enteric neurogenesis by inhibiting histone deacetylase activity; GPR41 and GPR43 receptors are expressed by epithelial cells and enteric neurons and are activated by short-chain fatty acids; bacteria-produced hormone-like molecules and neurotransmitters activate both, intestinal ImS and ENS, e.g., serotonin (5-HT) production by intestinal epithelium enterochromaffin cells [56].

Evidence has suggested that microbiota plays a crucial role in brain–intestine-ImS communication [56,61,62] through the modulation of VN activity [63]. This is demonstrated by experiments in which ingestion of the probiotic bacterium Lactobacillus rhamnosus (JB1) causes extensive neurochemical changes in the brain, reduces anxious behavior, or increases the recording of the mesenteric bundle that innervates the jejunum, these effects being abolished by vagotomy [64]. Although the mechanisms are unknown, vagal sensory neurons distinguish bacterial signals and transmit information to the brain by activating different brain nuclei and generating opposite effects on behavior. Evidence has demonstrated that ex vivo exposure of a colon section to peptidoglycan (main cell wall component of gram-positive bacteria) but not lipopolysaccharide (LPS, main cell wall component of gram-negative bacteria) evoked firing of vagal nerve endings [65,66] and excited vagal sensory neurons through the PAR-2 and TLR-4 receptors, respectively [63].

Gut inflammation impairs the epithelial barrier (leaky gut), enabling the translocation of microbiota, food particles, or toxins and leading to an immune response characterized by cytokine production, mast cell, and macrophage activation [4]. These inflammatory inputs modulate ImS that, in turn, detect and modulate the inflammatory processes [7]. Moreover, inflammatory mediators and gut products activate VN fibers that may affect ImS by pro-inflammatory adrenergic stimulation and anti-inflammatory cholinergic inhibition of cytokine production in macrophages and neutrophils [67]. This mechanism is, at least in part, under the cortical influence and can be indirectly measured as heart rate variability, an indirect measure of the activity of the cardiovagal VN branch associated with an individual’s stress regulation capacity [68,69]. Individuals with lower stress regulation capacity have an increased adrenergic stimulation and, through the CNS/ImS interaction, could increase the production and release of macrophagic cytokines [70,71] that trigger the NF-κB pathway [72,73] and cause an increase in reactive oxygen species (ROS) production [74,75]. Increased ROS due to reduced capacity to regulate psychological stress can eventually damage DNA telomeres [76,77], resulting in cellular senescence [67,78,79]. Evidence in IBD suggests an aberrant neural regulation of gastrointestinal tract submucosal arterioles [80] that do not vasodilate in response to parasympathetic acetylcholine due to changes in the endothelial release of vasodilators [81], also hypothesizing changes in the vascular tone during inflammatory diseases of the gastrointestinal tract.

## 5. Efferent and Afferent Innervation of the Gut

### 5.1. Anatomo-Functional Organization of the Vagus Nerve

VN is the tenth cranial nerve and has the longest pathway with most territories outside the head [82]. VN originates bilaterally in the medulla with many small rootlets converging in one trunk that exits the skull at the level of the jugular foramina, where lie two ganglia, the superior (jugular) and inferior (nodose). Recurrent branches penetrate the skull and innervate the meninges of the posterior cranial fossa. 

From the jugular ganglion and within the jugular foramen, VN also gives off the auricular branch, also termed Arnold’s nerve or Alderman’s nerve, that provides somatosensory innervation to the external ear [83]. The auricular branch transversely passes through the facial canal, entering the small canal of the petrous bone, and emerges from the tympanomastoid fissure, proceeding to innervate the external acoustic meatus and auricle [84]. The auricular branch of the VN is most prominently spread through the antihelix, tragus, cymba concha, and concha [85]. These zones are largely innervated by afferent A-fibers [86] that project mainly in STN and, to a lesser extent, in the spinal trigeminal nucleus [87]. However, the auricular branch of the VN terminals partially overlaps with the terminals of at least three other nerves, the auriculotemporal nerve (mandibular nerve, V3), the greater auricular nerve, and the minor occipital nerve [88]. The auriculotemporal nerve innervates the tragus and the anterior regions of the ear and supplies parasympathetic fibers to the parotid gland [89,90]. The great auricular nerve innervates both surfaces of the outer ear, while the minor occipital innervates the posterior surface, as well as the margins of the anterior part of the external ear [88]. The auricular branch of the VN underlies the cough reflex in humans after irritation of the posterior wall of the external acoustic meatus, but also other somewhat unusual somato-visceral reflexes, including (i) the gastro-auricular phenomenon, (ii) the pulmonal-auricular phenomenon, (iii) the auriculo-genital reflex, and (iv) the auriculo-uterine reflex [91].

Caudally from the nodose ganglion, the vagus nerve can be anatomically divided into three segments, the cervical, the thoracic, and the abdominal.

In the neck, the VN enters the carotid sheath that contains the common carotid artery and the internal jugular vein and gives off the pharyngeal branch for constrictors and most of the palatine muscles and the superior laryngeal nerve, providing sensory innervation of the laryngeal mucosa (internal branch) and motor innervation of the inferior constrictor of the pharynx (cricopharyngeal) and cricothyroid muscles (external branch). At the lower neck, VN gives off to the superior and middle cervical cardiac branches and the aortic branch (also known as the depressor nerve according to its role in lowering blood pressure), whose axons innervate the aortic arch on the left side and bifurcation of the brachiocephalic trunk on the right side.

In the thoracic region, VN enters the superior mediastinum, passing anteriorly to the aortic arch on the left and the subclavian artery on the right side and then behind the lung’s roots. Here VN forms two plexuses around the esophagus (the anterior is formed by the left VN, the posterior by the right VN) that pass through the diaphragm muscle and constitute the abdominal VN. During its pathway along the mediastinum, VN gives off the inferior cervical cardiac branch, the thoracic cardiac branches, the inferior laryngeal nerve (recurrent nerve), and the pulmonary plexus.

VN enters the abdominal cavity with the esophagus through its hiatus and with an anterior/posterior disposition, giving off branches that innervate the entire gastrointestinal tract until the splenic flexure of the colon. The posterior abdominal VN gives rise to the criminal nerve of Grassi, along the lesser curvature, and the dorsal coeliac branch that converges with the ventral coeliac branch of the anterior abdominal VN into the coeliac ganglion, from which vagal fibers depart to the intestines, pancreas, and potentially the spleen. The anterior trunk of the VN, moreover, gives off the common hepatic branch for the liver and bile ducts and completes the innervation of the stomach along the lesser and greater curvatures, the pyloric sphincter, the pancreas, and the proximal duodenum.

According to the nerve fiber composition, VN is a mixed nerve, with both sensory (80–90%) and motor (including postganglionic parasympathetic) (10–20%) fibers, distinguished in afferent general somatic, afferent general and special visceral, general and special visceral efferent axons (Figure 4). Most afferent fibers terminate into the central and caudal segments of the STN of the brainstem with a somatotopic pattern (the cardiovascular afferents project principally to the dorsal subnucleus, the pulmonary afferents to the ventral subnucleus while the caudal commissural subnucleus receives afferents from all visceral components [92]). Among the others, these general visceral afferent fibers carry information related to the environment and functional state of the viscera. 

Efferent fibers of the VN originate from both the ventral motor nucleus (nucleus ambiguous) and the dorsal motor nucleus and provide the innervation of the pharyngo-laryngeal muscles and parasympathetic innervation of the neck, thoracic, and abdominal organs up to the splenic flexure of the colon, respectively [93].

### 5.2. VN Innervation of Gastrointestinal Tract

The different anatomical and molecular profiles of the VN fibers innervating the gastrointestinal tract constitute the substrate for the regulation of its main functions, including feeding behavior, digestion, and metabolism. Vagal afferent axons form different kinds of sensory endings along the gastrointestinal tube that allow the detection of chemical and mechanical stimuli. VN A-type sensory fibers present intraganglionic laminar endings (IGLE), tension slowly/rapidly adapting receptors for control of muscle distension, or mucosal and muscular-mucosal receptors, sensitive to muscle distension and mucosal deformation. Mucosal endings are largely C-fibers terminating in the lamina propria of the gastric and intestinal mucosa and allow the detection of chemical stimuli from the lumen (several micronutrients, hormones, and pH changes) but also mechanical stimulation. The distribution of mucosal endings changes from the stomach to the intestine, being more expressed in the proximal small intestine. In the small intestine, mucosal endings do not make contact with the intestinal lumen and communicate with second-order chemosensory cells, the enterochromaffin cells, the principal source of serotonin in the body, forming neuropod-mediated synapses.

Enterochromaffin cells of the gastrointestinal tract include the enteroendocrine cells of the gut epithelium [94]. These cells contain secretory granules that are believed to contain peptide hormones (CCK, peptide YY, glucagon-like peptides) but also neurotransmitters (in particular glutamate) and present an axon-like process (called neuropod) that passes into the submucosa and contacts enteric glial cells and afferent vagal fibers [94]. Experimental evidence has demonstrated that most of the CCK-positive enteroendocrine cells form a glutamatergic synapse at the level of their neuropod to afferent vagal fibers, creating a neuroepithelial circuit that transduces gut luminal signals. Therefore, enteroendocrine cells receive signals from the gut mucosa, food components, drugs, and microbiota, stimulating the secretion of hormones (that regulate intestinal secretion and motility, facilitating the ingestion, digestion, and absorption of nutrients) but also transmit sensory signals from the gut to the brain and receive efferent signals from the brain [95]. 

Innervation of the muscular layers of the gastroenteric tube is provided by intramuscular arrays, consisting of branching fibers in the circular and longitudinal muscular sheath acting as stretch receptors. These fibers are mainly unmyelinated and respond to gastrointestinal distension and chemical stimuli (gut hormones). Muscular-mucosal receptors of vagal endings lie in the lamina propria layer, creating a subepithelial plexus that responds to both light mucosal stroking and distension. These receptors are sensitive to harmless mechanical stimuli in inflammatory conditions that sensitize and trigger mechano-sensitivity in muscular-mucosal and vascular afferents.

Finally, the strict link between vagal afferent fibers and ImS modulation has been recently investigated, showing that cytokine-specific information is present in sensory neural signals within the vagus nerve [96].

Therefore, VN and the dorsal vagal complex, through the organization of vagovagal reflexes, establish interconnections between the entire neuroaxis of the CNS and the gut and is responsible for a bidirectional communication pathway between the intestine and brain [97].

### 5.3. Sympathetic Innervation

Sympathetic innervation of the GI tract drives noradrenergic (NA) fibers from the prevertebral ganglia and splanchnic nerves to the enteric tube smooth muscle wall, intramucosal/submucosal plexus ganglia, and arteries [98,99]. In particular, the celiac-mesenteric ganglia provide fibers to the stomach, small intestine, and proximal large intestine, the inferior mesenteric ganglia to the intermediate large intestine, and pelvic ganglia to the distal part of the large intestine (rectum). 

In the gastrointestinal tract, sympathetic postganglionic neurons provide vasoconstrictor input to arterioles while enteric submucosal vasomotor neurons and afferent neurons innervate the vessels and can lead to their vasodilatation [100,101]. In particular, the sympathetic nervous system can pronounce vasoconstriction and is influenced by factors regulating vascular function (NO, ROS, endothelin, and the renin-angiotensin system). At the microcirculation level, angiotensin-II facilitates neuronal transmission within sympathetic ganglia [102,103], favors norepinephrine release by sympathetic nerve terminals, acting on pre-synaptic receptors and increasing vasoconstriction in arterioles [103,104].

In experimental setups, NA fibers of the enteric nervous system are marked using antibodies against tyrosine hydroxylase (TH), the rate-limiting enzyme in the biosynthesis of dopamine and NA. Evidence has shown a direct and inverse correlation between age and the TH staining in the circular smooth muscle [105].

Sympathetic nerves also provide an important role in intestinal inflammatory response through their connections with the cholinergic anti-inflammatory pathway (CAP). The relevance of the sympathetic system to intestinal inflammatory mechanisms derives from patients with IBD, showing an enhanced sympathetic tone. Accordingly, experimental evidence has demonstrated that NA depletion using the reserpine, an irreversible reuptake-inhibitor of VMAT-2 (vesicular monoamine transporter-2), prevented the anti-inflammatory action of vagal stimulation in rats [106], thus resulting in detrimental. Interestingly, sympathectomy (rather than vagotomy) has opposite effects on inflammatory mechanisms in experimental colitis whilst electrical stimulation of superior mesenteric ganglia reduced gut inflammation [107]. These conflicting results might be related to variable expression of adrenergic receptor subtypes (β2 and β3 vs. α2) in the intestinal inflammatory process and catecholamine control of the intestinal microbiota composition [107,108,109,110]. 

The effects of the sympathetic system on the inflammatory mechanism and their cross-interaction with the counterpart parasympathetic need to be further investigated. Contrasting results of stimulation protocols need to be interpreted in light of the different experimental setups for their ability to enhance or reduce the inflammatory cascade.

## 6. Immune-Mediated Process and Inflammation in Crohn’s Disease

### 6.1. Crohn’s Disease

Crohn’s disease (CD) is an autoimmune disorder subsequent to an inappropriate innate and acquired immune response to harmless antigens that can normally pass through the gastrointestinal tract. This disease is characterized by inflammation of the entire gastrointestinal wall and can be located anywhere from the mouth to the anus. Although the origin is unidentified, it seems to be the consequence of a combination of genetic and environmental factors that alter the immune response in the gastrointestinal tract [5]. The risk of CD and other inflammatory bowel disorders has been correlated to antibiotics exposure (in particular penicillin/cephalosporins) that, altering the intestinal microflora composition, might trigger an aberrant immune response and an inflammatory cascade [111]. 

At the beginning of the disease, the intestinal epithelial barrier is damaged, whereby the antigens existing in the intestinal lumen penetrate the epithelium. Specific ImS cells (macrophages, dendritic cells) can recognize these antigens and trigger an immune response, producing pro-inflammatory mediators that chemo-attract other immune cells to the site of inflammation and induce differentiation of undifferentiated T cells. T cells can differentiate into Th1 or Th17 cells, producing more inflammatory mediators and sustaining inflammation and damage to the intestinal epithelium. When these processes are maintained, inflammation becomes chronic, and it can generate local and systemic complications [5,112].

Current treatment of CD is based on the use of corticosteroids, aminosalicylates, thiopurine, and immunomodulatory and biological drugs [113]. Sulfalazine, masalazine, and glucocorticosteroids have proven positive effects on intestinal inflammation, showing clinical efficacy and high efficacy in remission induction. Immunosuppressants are usually indicated for maintenance therapy to spare corticosteroids whilst biological therapy, based on antibody activity, modulates the inflammatory mechanisms blocking its mediators, especially TNF-α [114].

This wide pharmacological approach, however, not always provides positive effects, hypothesizing that other factors are involved and reduce the drug efficacy. Among others, the diet is able to change intestinal barrier function, affect the structure and function of intestinal flora, and promote immune disorder, thus promoting inflammation [115]. Therefore, diet control and nutraceutical product consumption might be supportive of the primary therapy for their ability to modulate the inflammatory process. 

A non-pharmacological therapy that is increasing in importance is treatment with polyphenols. Phenolic compounds or polyphenols are secondary metabolites with a wide presence in the human diet, such as fruits, vegetables, and products derived from commonly consumed plants such as cocoa, tea, or wine. These compounds have demonstrated through in vitro, ex vivo, and animal tests, different beneficial effects on health due to their antioxidant, anti-inflammatory, estrogenic, cardioprotective, cancer chemopreventive, and neuroprotective properties, acting either directly when absorbed or indirectly, after being processed by the intestinal microbiota [116,117,118,119,120,121]. These compounds also exert their importance by modulating the composition of the microbiota. They promote human health through a balance in intestinal microbes, stimulating the growth of beneficial bacteria (*Bifidobacteria*, *Lactobacilli*, *Actinomycetes*) and inhibiting pathogenic bacteria (*Escherichia, Enterococcus, Campilobacter, Clostridium*), exerting effects similar to prebiotics [116,117,119,122,123].

The human body identifies ingested polyphenols as xenobiotics making them less available than micro- and macro-nutrients. The majority of these polyphenols, those with oligomeric and polymeric forms of high molecular weight, reach the colon, where they are broken down by intestinal microbiota and transformed into absorbable metabolites with biological activity; a small portion of these polyphenols, those with monomeric and dimeric structures, are absorbed in the small intestine [116,118,124]. Phenolic compounds are also obtained by gut bacteria hydrolysis of glycosides, glucuronides, sulfates, amides, esters, and lactones, but also by ring cleavage, reduction, decarboxylation, demethylation and dehydroxylation reactions [119,122]. If absorbed, these phenols are transported to the liver through the portal vein, and there they are further processed before entering the systemic circulation and distributed to the whole body or eliminated by the urine [116,122]. A part can be recycled back to the small intestine in bile or by transporters, reducing bioavailability [118,122]. 

The low final concentration of polyphenols in plasma suggests that direct antioxidant effects would be expected only in the tissues that are directly exposed to these molecules (e.g., the gastrointestinal tract), although they could also accumulate in some compartments interacting with other membranes or that they could deconjugate to reach different tissues [122]. The degree of degradation of the compounds, and therefore, bioavailability and bioefficacy, is highly influenced by the concentration of substrate in the diet and the variations of the colonic microbiota [116,118,119,122,124,125].

Polyphenols affect the composition of the intestinal microbiota with their prebiotic activity while providing precursors for microbes-produced metabolites that protect the intestinal immunomodulation barrier [126]. In vitro studies have sown polyphenols inhibit oxidation of low-density lipoproteins (LDL) [117], regulate cholesterol-related processes, limit lipid production by decreasing lipogenesis and adipogenesis and stimulating lipolysis [124], inhibit cancer cell lines growth by slowing down telomerase, lipoxygenase, and cyclooxygenase activity, and also interact in different signal transduction pathways, in cell cycle regulation, platelet function and in preventing endothelial dysfunctions [117]. In vitro and in vivo studies have shown that epigallocatechin-3-gallate (EGCG), genistein, curcumin, resveratrol, genistein, pomegranate extract, fisetin, nobiletin, and lupeol have chemopreventive and/or chemotherapeutic effects against prostate cancer by inducing apoptosis in tumor cells [124,127]. Clinically, it has been observed that flavonol intake can reduce systolic blood pressure and plasma triglyceride concentration [120]. 

Of particular interest are the microbiota–gut–brain axis and the modulatory activity of the vagus nerve. Brain behavior and mental health are indirectly modulated by polyphenols through this axis in a bi-directional direction in which, in addition to the vagus nerve, also participate the immune system, neuroendocrine pathways, and bacteria-derived metabolites [128] (Figure 5). 

Microbiota dysregulation is linked to neuroinflammatory and neuroimmune diseases (Lombardi et al., 2018), so polyphenols’ action on the microbiota (either directly or through the byproducts of the metabolism of the bacteria) could modulate oxidative stress, inflammation, and neurotransmitter imbalance through the activation of the hypothalamus–pituitary–adrenal axis (HPA axis).

In vitro and in vivo results suggest that both food polyphenols and their microbial metabolites contribute to the effect of these molecules on host health [116,119,120,121,124]. However, the complexity of these studies lies in the bidirectional relationship between microbiota and polyphenols, with microbiota being responsible for metabolizing them while they modulate the microbial composition through selective prebiotic effects and antimicrobial activities against intestinal pathogens [116,118]. Techniques to monitor bacteria and their activity, as well as to identify and quantify flavonoids in foods and plants, along with their metabolites, are necessary to improve our knowledge in this field [119,124]. Combining metagenomic and metabolomic studies will allow a greater understanding of the interaction between microbiota and dietary phenols, allowing the prevention of intestinal diseases, such as inflammatory bowel diseases or colon cancer, as well as diseases in other tissues [119].

### 6.2. The Innate Immune Response into Crohn’s Disease

The innate immune response acts as the first line of defense in the gastrointestinal tract. In Crohn’s disease, the innate immune response in the gastrointestinal tract is triggered by (1) the increased permeability of the intestinal epithelium, (2) the dysfunction of Paneth cells, and/or (3) the decreased tolerance of dendritic cells and macrophages to intestinal lumen antigens [5,129] (Figure 6).
The intestinal epithelium normally blocks the paracellular passage of antigens present in the intestinal lumen through junctions between intestinal epithelial cells [5]. In Crohn’s disease, an abnormal increase in permeability allows the paracellular passage of antigens to the lamina propria, lying beneath the epithelium, in which resident macrophages and dendritic cells can identify antigens and present them to T cells, triggering an inflammatory response. In a high percentage of patients with Crohn’s disease, the increased permeability of the intestinal epithelium is produced by mutations in the NOD2 gene. Mutations in this gene allow antigens within the gastrointestinal tract to spread through intercellular spaces to the lamina propria and trigger inflammation [129].Paneth cells are highly specialized intestinal epithelial cells that excrete granules with antimicrobial peptides, such as α-defensins, and control the commensal microbiota [130]. Several studies have linked Crohn’s disease with dysfunctional Paneth cells, which generate dysmorphic and functionally altered granules, producing inflammation in the ileum [5,131]. Furthermore, the abnormal activity of Paneth cells can induce stress in their endoplasmic reticulum, which is correlated to autophagy and to unfolded protein response [1,5,132].In the absence of disease, ImS does not trigger immune responses to many antigens (tolerance). In Crohn’s disease, the immunological tolerance of dendritic cells and/or macrophages could be diminished, and once they recognize antigens, they polarize undifferentiated T lymphocytes (naive T lymphocytes) to differentiate into Th1 or Th17 by means of specific inflammatory mediators: IL-12 or IL-23, respectively [133]. This can increase the amount of other inflammatory mediators (IL-17) and/or induce the proliferation of differentiated immune cells such as Th17 lymphocytes [134]. IL-17, in turn, induces epithelial cells secretion of IL-8, a pro-inflammatory mediator that boosts the immune response by promoting the recruitment of neutrophils in the inflamed intestine, whereas the recruitment of neutrophils could be under the control of the neuro-immune response [135]. Additionally, there is evidence that Th17 cells are further amplified by IL-21, which controls IL-17 secretion by lymphocytes located in the lamina propria [134,136].

Finally, another factor to consider is dysbiosis, understood as an unfavorable alteration of the composition and function of the intestinal microbiota. The increase in pathogenic bacteria with the ability to adhere to the intestinal epithelium can affect the permeability of the intestine, also altering the diversity and composition of the microbiota and inducing inflammatory responses. Although a direct causal relationship has not been established, many studies have reported that the composition of the microbiota in IBD patients is altered compared to healthy subjects [58,60].

### 6.3. The Acquired Immune Response into Crohn’s Disease

The acquired immune response is also dysregulated in Chron’s disease and characterized by (1) an imbalance in the number of lymphocytes, (2) an increase of Th1 cytokines, and (3) an increase of lymphocytes in the mucosal lamina propria [5] (Figure 6). 

As mentioned above, macrophages and/or dendritic cells participate in the differentiation of naive T lymphocytes into Th1 or Th17 lymphocytes. If this process becomes chronic, an imbalance between effector T cells (Th1 and/or Th17 lymphocytes) and regulatory T cells can appear [137]. IL-12 can stimulate Th1 cells to secrete specific inflammatory mediators (TNFα, interferon γ, and IL-2), which sustain inflammation and enable the ImS to preserve the mucosa from antigens [137,138]. Th17 lymphocytes also maintain inflammation with IL-17 and IL-22 through a specific nuclear receptor called retinoic acid receptor-related orphan receptor (ROR) [138]. In contrast, regulatory T cells (Treg) secrete mediators with an anti-inflammatory effect, such as IL-10 and transforming growth factor β (TGF-β) [134]. Several studies have described a disproportion between genes promoting programmed cell death (pro-apoptotic) and those that inhibit it (anti-apoptotic), resulting in a decrease of T cells apoptosis, increasing their survival [139] and a release of pro-inflammatory cytokines that damage the intestinal mucosa. This process, in addition to the lymphocytes increase caused by the presentation of antigens by macrophages, significantly increases the number of lymphocytes scattered in the lamina propria [140]. 

After the polarization of Th1 lymphocytes from naive T lymphocytes, Th1 produces a great amount of cytokines, tumor necrosis factor-α (TNF-α), interferon-γ (IFN-γ), and IL-2 that increase even more when these cells are stimulated by the inflammatory mediator IL-12 [134]. The cytokines released by Th1 lymphocytes promote not only the recruitment of inflammatory cells but also the death of resident cell populations and the damage of the tissue [134]. 

Alterations of innate and adaptive immune responses are currently correlated to persistent bowel inflammation and also dysfunction of the ENS and gut–brain axis [141,142]. Therefore, ENS should be considered as a target for treating IBD [6]. To date, Crohn’s disease has no definitive cure, but there are disease-modifying drugs and treatments that control the symptoms by reducing the immune response or inflammation, including mesalazine, corticosteroids, and monoclonal antibodies or bioengineered receptors targeting inflammatory cytokines, such as TNFα (etanercept), IFN-γ, or preventing lymphocytes migration (natalizumab, vedolizumab), or IL-12/IL-23 blocking the onset of the inflammatory cascade triggered by Th1 and Th17 lymphocytes (ustekinumab) [129,143,144]. However, these therapeutic options are not always effective, and a large proportion of patients become refractory to available treatment options [145].

Therefore, non-pharmacological therapies seem to be of interest in IBD treatment and are popular among patients: neuromodulation, based on the bidirectional interaction between CNS and ImS, VNS being the most likely therapeutic procedure [146,147].

## 7. Targeting Inflammation with Vagus Nerve Stimulation

To date, electrical left cervical VNS has been approved and used for the treatment of refractory epilepsy since 1997, treatment-resistant depression since 2005, and cluster headaches since 2016 [91,148]. Transcutaneous electrical stimulation of the ABVN obtained EU approval for the treatment of refractory epilepsy and pain [149].

The stimulation parameters, which are typically in the range of 10–20 Hz, 0.25–2 mA intensity, and 0.1–0.5 ms pulse width, are optimized to activate mainly vagal afferent fibers [150]. Activation of vagal afferent fibers has been verified in VN-brain-related areas using fMRI, showing activation of the left locus coeruleus, thalamus, insula, prefrontal cortex, posterior cingulate gyrus, and the bilateral postcentral gyrus, and deactivation in the right nucleus accumbens and cerebellar hemisphere [151,152].

This extensive neural network, also involving the sympathetic nervous system and neuroendocrine systems, together with local microglia and astrocytes, underlies appropriate behavioral, autonomic, neuroendocrine, and inflammatory responses.

The detection of the inflammatory mediators in the body allows precise identification of the inflammation site. As previously described, the innate immune response is activated by the detection of an antigen in the body, which causes the release of inflammatory mediators in the region of detection. Inflammatory mediators, in turn, activate VN fiber endings, which generate postsynaptic excitatory potentials into STN neurons according to a somatotopic pattern [32,153]. Activation of the specific STN subnucleus by the inflammatory process stimulates two different mechanisms via the VN path: (1) the CAP and (2) the hypothalamic–pituitary–adrenal axis (HPA) [154,155,156]. These mechanisms are part of the central-autonomic-network output that could be regulated by the medial prefrontal cortex due to its descending connections to pre-autonomic cell groups in the hypothalamus, periaqueductal gray, and brainstem [157].
***The CAP***: STN, activated by general visceral efferent fibers of VN, project to the dorsal motor nucleus of the vagus (DMx), i.e., the parasympathetic nucleus of the VN, that synapses in celiac ganglia and sends sympathetic fibers to the spleen through the splenic nerve (Figure 7).Therefore, the spleen, as a key component of the cholinergic anti-inflammatory pathway, could provide a potential therapeutic target for immune-mediated diseases [158]. Splenic nerve endings in the spleen activate T lymphocytes expressing acetylcholine transferase (ChAT + T cells). When the traveling T cells identify macrophages, they secrete acetylcholine (ACh) that inhibits the release of macrophages inflammatory mediators [159] through (1) downregulation of nuclear translocation of NF-kB and (2) activating Janus kinase (JAK)/signal transducer and activator of transcription in macrophages and other cytokine-producing cells via nicotinic acetylcholine receptors (nAChR), which leads to suppression of proinflammatory cytokines [160,161,162].The importance of the spleen in the cholinergic anti-inflammatory pathway and in mediating the protective effect of efferent VNS was reported in an animal model in which the anti-inflammatory effect of VNS was abolished in animals lacking the α7 subunit of the nAChR (α7nAChR) due to splenectomy [159,163]. Moreover, an adoptive transfer of VNS-conditioned α7nAChR splenocytes conferred protection to recipient mice with ischemia-reperfusion injury (IRI) [161]. However, only recently the role of the spleen and α7nAChR has been proven for IBD [164,165]. In IBD, a decreased VN activity is observed, and it has been reported that central cholinergic activation in mice models of IBD might result in reduced mucosal inflammation. This is associated with decreased major histocompatibility complex II (MHC II) levels and pro-inflammatory cytokine secretion by splenic CD11c⁺ cells mediated by α7nAChR signaling. This anti-inflammatory cholinergic-mediated effect was abolished in mice with vagotomy, splenic neurectomy, or splenectomy [166]. The key role of the spleen has also become more intricate due to the discovery of the splenic glial cells, which may represent active partners mediating immune response [167]. Likely other glial cells, splenic glial cells are elongated cells of the white pulp expressing GFAP and S100β that ensheath and support both sympathetic and sympathetic fibers, guiding axons during outgrowth and response to injury and disease [168,169,170,171,172]. Moreover, through an extensive network, splenic glial cells make contact with lymphocytes, thus mediating the neuro–immune interaction. It is conceivable that adrenergic signaling of sympathetic fibers modulates immune reaction via splenic glial cells that express both adrenergic and purinergic receptors [173].***The hypothalamic–pituitary–adrenal axis***: the cholinergic anti-inflammatory pathway also involves a response of the hypothalamic–pituitary–adrenal axis (Figure 7). VN inputs to STN modulate the membrane potential of noradrenergic neurons within STN (group A2) that project to the parvicellular neurons of the paraventricular hypothalamic area (PVH) that, in turn, release a specific hormone, the corticotropin-stimulating factor (CRF) [174]. CRF binds to specific receptors expressed by cells of the pituitary gland, releasing adrenocorticotropin, a hormone that modulates the cells in the zona fasciculata in the adrenal glands that, in turn, release glucocorticoids, which play a strong anti-inflammatory role [153,155,156].

Several experimental and clinical studies have described an anti-inflammatory effect associated with VNS. A reduced release of inflammatory cytokines (IL-6, IL-1β, and TNF-α) was reported in an animal model with behavioral deficits induced by ischemic/reperfusion cerebral deficits and treated with VNS [175]. The decrease of pro-inflammatory cytokines generates an anti-inflammatory and anti-apoptotic effect which in turn can reduce behavioral deficits [175]. Since many diseases involve different inflammatory pathways, VNS could be used as a treatment for diseases that involve chronic inflammatory diseases, such as Crohn’s disease, ulcerative colitis, or rheumatoid arthritis. VNS activates cholinergic receptors of VN, both α7nAChR and muscarinic AChR3 (M3AChR), resulting in the activation of the cholinergic anti-inflammatory pathway [176]. The activation of cholinergic receptors reduces TNF-α, IL-6, and IL-1β release and modulates the inflammatory response [176]. These pro-inflammatory cytokines are also reduced by activating the peroxisome proliferator-activated receptor (PPAR) receptor [177], distributed in some regions innervated by the afferent VN such as the hippocampus, the hypothalamus, the basal ganglia, the thalamus, and the striatum [178]. Moreover, VNS downregulates the translocation of NF-κB and myosin light chain kinase, which could attenuate the disruption of the intestinal epithelial barrier in IBD [179]. VNS also reduces the pain related to IBD, promoting a gate mechanism in the spinal cord, activating the ascending pain inhibitor systems, and through the release of neurotransmitters such as endorphins, dynorphins, substance P, and other endogenous opioids with analgesic effects [180].

## 8. Conclusions and Future Perspectives

Targeting the inflammatory process of IBD with artificial stimulation of certain branches of VN could be achieved through neuroprostheses to restore intestinal homeostasis. The main disadvantages of stimulating the cervical VN are the highly invasive approach and the likely stimulation of nerve fibers that are not in the cervical trunk of the VN, which can cause adverse side effects [181]. For this reason, the stimulation of the auricular branch of the VN may represent an effective and non-invasive alternative.

Neuroprostheses are a novel therapeutical tool that can be used to restore specific neural functions [182]. Recent studies have shown that this approach could be more robust and natural compared with previous strategies [183]. The ideal neuroprosthetic interface allows neural recording and stimulation of selected areas of the nervous system while reliably providing clinical benefits over long periods [182]. Current neuroprosthetic research has yielded a host of new technologies that can promote neural regeneration, enable communication, and restore mobility post-injury [182]. Therefore, this approach could also be applied to a wide range of chronic diseases that involve an altered neural function in their physiopathology.

Finally, a matter still to be clarified is the effect of VNS on the intestinal microbiota, either beneficial or harmful, which are the changes induced to the resident microbiota and, consequently, its contribution in triggering other chronic diseases or in modulating the effect of treatment. Inversely, VNS’s direct and indirect beneficial role should also be elucidated. Moreover, since the employment of VNS as a therapeutic strategy, either “per se” or as a complementary treatment for a multitude of disorders, it is essential to understand the secondary effects of this treatment on gut microbiota and the possibility of a consequent triggering (or facilitation) of the development of IBD-like diseases.

## Figures and Tables

**Figure 1 cells-11-04103-f001:**
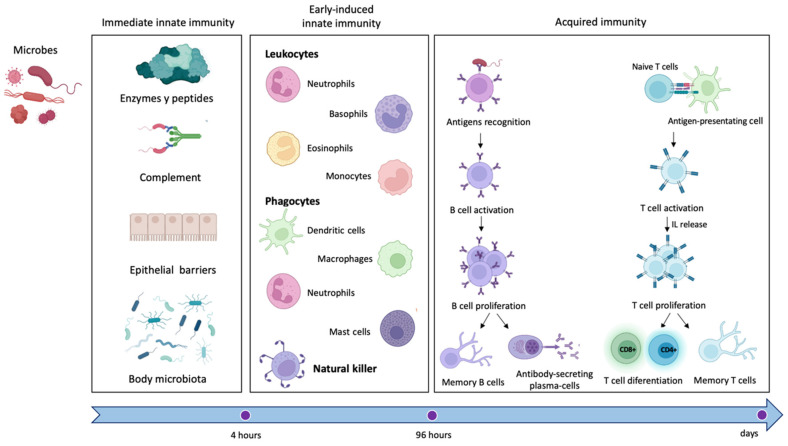
Immune response. Schematic representation of the key components of the innate (immediate and early induced) and the acquired response of the immune system. Immediate innate immunity is based on (1) the activity of circulating enzymes and peptides with anti-microbic action, including complement, (2) the integrity of the anatomic epithelia, and (3) the antagonism of resident host microbiota. Lately, early induced innate immunity is cell-dependent, modulated by the activity of leukocytes, phagocytes, and natural killer cells. Finally, adaptive immunity is lymphocyte-dependent: antigen is presented to B/T lymphocytes by antigen-presenting cells, which become active, proliferate, and differentiate into B cells producing antibodies, T cells CD4+ (T helper) or CD8+ (CTL, cytotoxic T lymphocytes) and B/T memory cells.

**Figure 2 cells-11-04103-f002:**
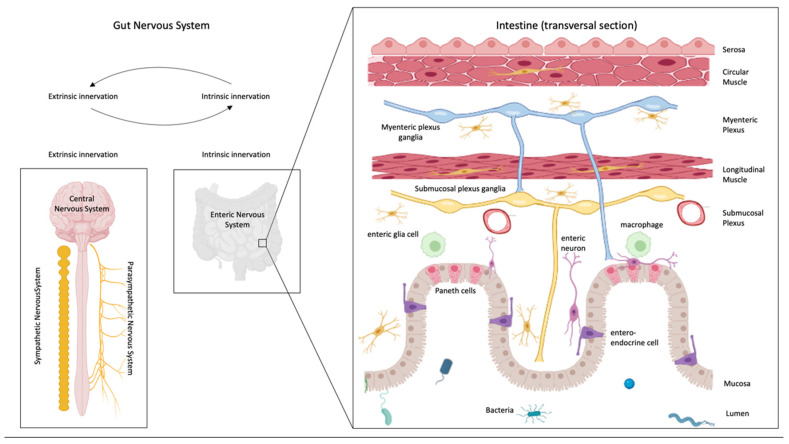
Gut nervous system. Intestine is heavily innervated by an intrinsic dense neural network (intrinsic nervous system) and by an extrinsic nervous system formed by the sensory and motor nerves of the central, sympathetic, and parasympathetic nervous systems. The enteric nervous system is composed of two complex neural plexuses, the myenteric, lying between the two muscles of the intestine, and the submucosal, placed between the longitudinal muscle and the mucosa. There is bidirectional communication between the intrinsic and the extrinsic system, and both of them communicate/interact with bacteria and immune system cells.

**Figure 3 cells-11-04103-f003:**
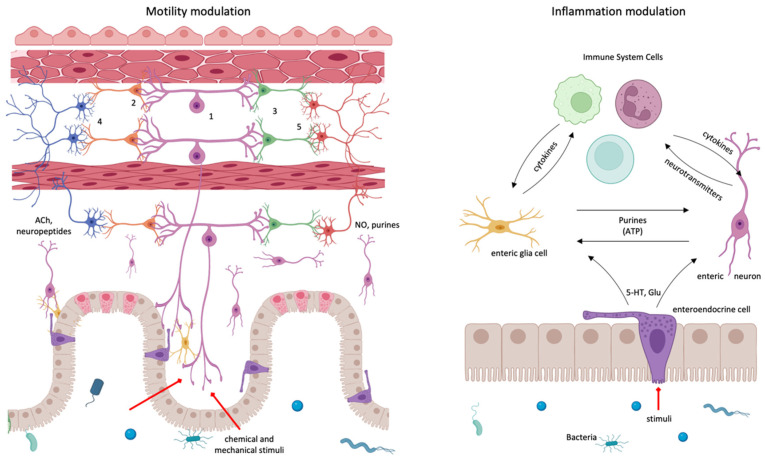
The enteric nervous system is mainly devoted to the modulation of the motility of the intestine (left) and to the modulation of the inflammatory processes that take place in this organ (right). In motility regulation, chemical and mechanical stimuli excite IPAN, intrinsic primary afferent sensory neurons (1). IPAN synapse with proxima-ascending (2) and distal-descending (3) interneurons, which, in turn, activate excitatory (4) or inhibitory motoneurons (5). IPAN produces substance P and calcitonin gen-related peptide; interneurons employ several neurotransmitters, including ACh; excitatory motneurons are ACh-ergic or neuropeptidergic, and inhibitory motoneurons are purinergic, NO-producing, etc., neurons. Inflammation regulation is carried out thanks to a tri-partite unit formed by the intercommunicating mucosa, gut nervous system, and immune system. Communication between intestinal epithelium and neural system is performed through the mucosa enteroendocrine cells secreting peptidic hormones (CCK, YY peptide, glucagon-like peptides, etc.) and neurotransmitters, especially glutammate; immune system mainly employs cytokines; enteric glial cells respond to several neurotransmitters and other molecules but mainly responds to purines.

**Figure 4 cells-11-04103-f004:**
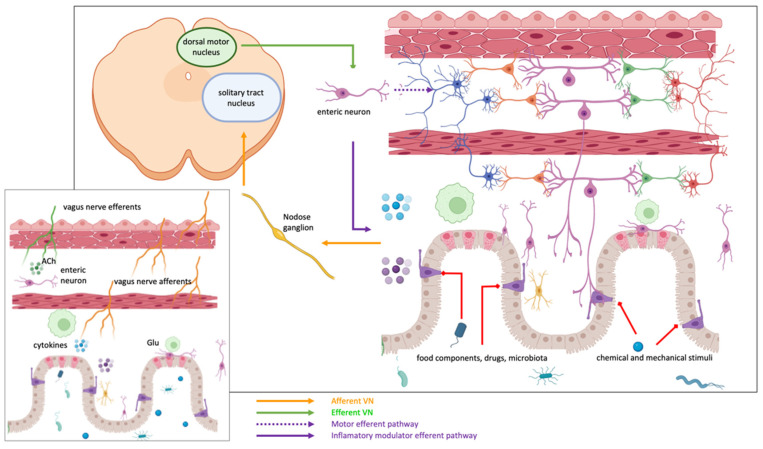
Vagus nerve afferent terminals are mechanoreceptors, chemoreceptors, and thermoreceptors. They are stimulated by muscular tension and stretching, hormones, nutrients, enteroendocrine cells’ neurotransmitters, and inflammatory mediators and are sensitive to osmolarity and pH changes. Their bodies are in the nodose ganglia and transmit information to the nucleus of the solitary tract. The efferent pathways reach the myenteric plexus, where they communicate with (1) enteric neurons that regulate motility and inflammation, (2) with cholinergic neurons that contact macrophages (anti-inflammatory pathway) or (3) with the sympathetic adrenergic system (inflammatory pathway).

**Figure 5 cells-11-04103-f005:**
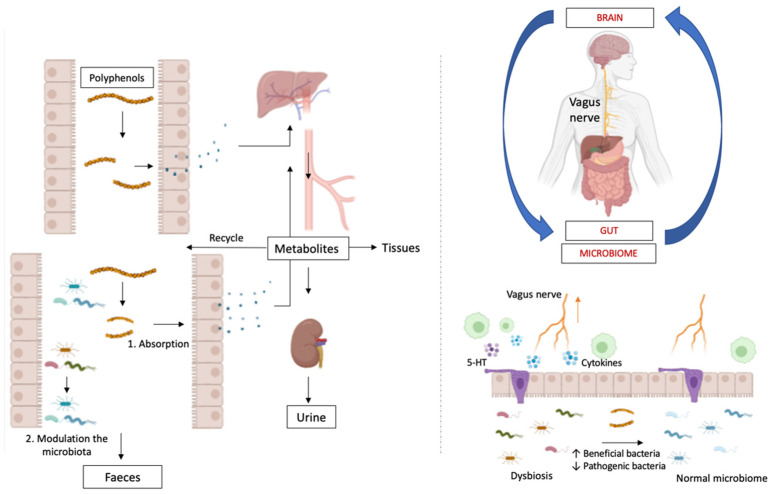
Once the polyphenols are ingested, a minimal part is metabolized in the small intestine and the majority in the large intestine. These are absorbed and passed to the liver, from where they can return to the intestine, be eliminated in the urine, or be distributed to different tissues or organs. In case of dysbiosis, polyphenols can help promote normal microbiota, thus reducing inflammatory mechanisms and sensitization of vagus nerve endings.

**Figure 6 cells-11-04103-f006:**
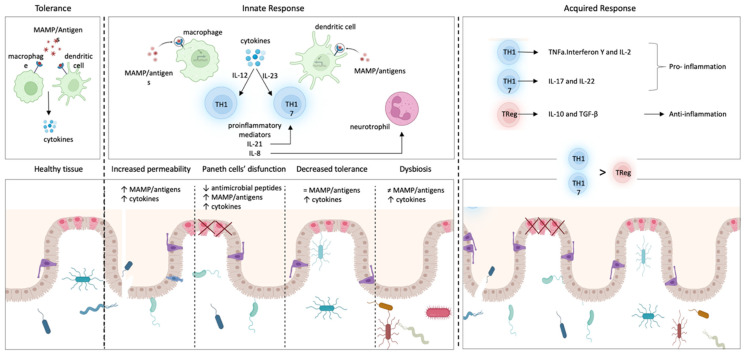
Immune-mediated process and inflammation in Crohn’s disease. The innate immune response acts as the first defense in the gastrointestinal tract. Specific cells of the immune system (macrophages, dendritic cells) can recognize these antigens and trigger an immune response, producing proinflammatory mediators that attract other immune cells to the site of inflammation and induce differentiation of undifferentiated T cells into Th1 or Th17. In Crohn’s disease, this response is triggered by: (1) increased permeability of the intestinal epithelium, (2) dysfunction of Paneth cells, which normally secrete antimicrobial peptides to maintain the microbiota, (3) decreased tolerance of dendritic cells and macrophages to antigens found in the intestinal lumen. Under normal conditions, the cells of the immune system communicate with the commensal microbiota through molecular recognition patterns (MAMPs) recognized by specialized receptors (4) dysbiosis of the gut microbiota. The acquired immune response is characterized by an imbalance of lymphocytes due to an increased number of Th cells in the mucosal lamina propria. Th lymphocytes generate pro-inflammatory molecules that, in turn, cause regulatory T lymphocytes to release anti-inflammatory factors.

**Figure 7 cells-11-04103-f007:**
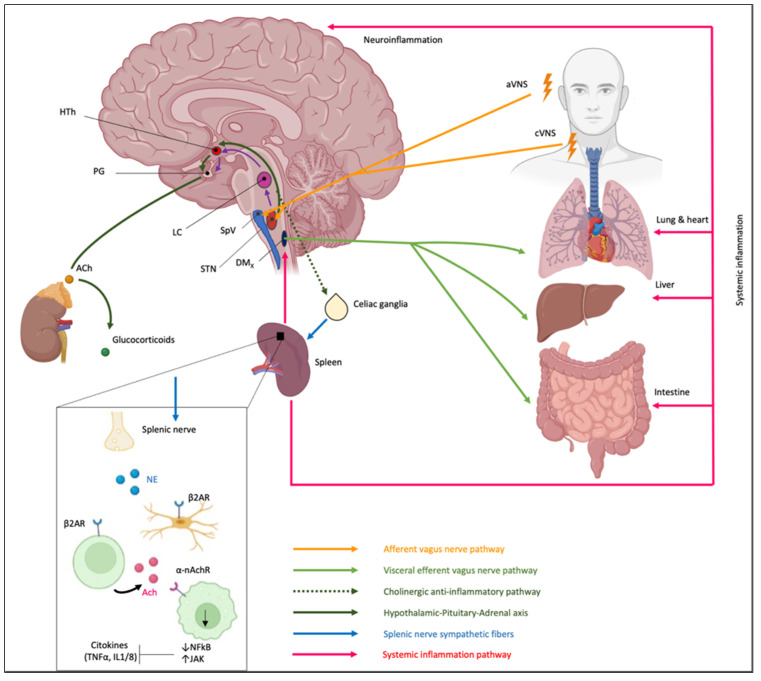
Vagus nerve stimulation-induced modulation of intestine immune responses. Artificial vagus nerve stimulation, both cervical (aVNS) and auricular (aVNS), directly excites brainstem relay neurons of the solitary tract nucleus (STN) and the spinal nucleus of the trigeminal complex (SpV), thus activating the visceral afferent, cholinergic anti-inflammatory and splenic nerve afferent pathways as well as the hypothalamic–pituitary–adrenal axis. DMx, dorsal motor nucleus of the vagus nerve; STN, solitary tract nucleus; SpV, spinal nucleus of the trigeminal complex; LC, locus coeruleus; HTh, hypothalamus; PG, pituitary gland.

## Data Availability

The datasets and materials generated and analyzed during the current study are available from the corresponding author upon reasonable request.

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
