# Peer review of "Vagus Nerve Stimulation: A Personalized Therapeutic Approach for Crohn’s and Other Inflammatory Bowel Diseases"

_cells, 2022, doi:10.3390/cells11244103_

Round 1

Reviewer 1 Report

Dear authors, 

     The article was carefully reviewed, and suggested major revisions.

The article by Cirillo et al., 2022 titled “Vagus nerve stimulation: a personalized therapeutic approach 2 for Crohn and other inflammatory bowel diseases” was carefully reviewed and suggested the following changes.

1. Typographical errors must be corrected.

2. Plagiarism report is needed.

3. There are some grammar mistakes that should be corrected.

4. There are several literature published on inflammatory bowel disease, how you justify this article is different from them?

5. Authors suggested the targeting therapeutic approaches against IBD. Available drugs have side effects therefore; scientific community looks forward the therapeutic options. Polyphenol regulated gut microbiota approach is a best strategy. Include literature on polyphenol approaches, these components contain immuno-modulatory and antioxidant properties. So, in this review, elaborate the polyphenol effect in therapeutic options against IBD.

6. Include tables which indicate the previous studies on this aspect and their possible therapeutic intervention studies.

7. Add another figure which shows the polyphenol induced gut microbiota modification in vagus nerve stimulation including its molecular mechanism.  

Author Response

Response to reviewers

Prof. Dr. Naweed I. Syed

MDPI - Cells

Section Editor-in-chief

Manuscript ID: cells-1908669

Dear Editor,

we have greatly appreciated the prompt and constructive revisions that our paper has received. We are now sending a revised version of the manuscript, having addressed the reviewers’ comments.

Therefore, we hope that the responses and the revised manuscript are satisfactory, and meet with your approval.

We look forward to your decision.

Your sincerely,

                                                                                             Giovanni Cirillo

Prof. Giovanni Cirillo, MD, PhD

Division of Human Anatomy, Neuronal Networks Morphology and Complex Systems Biology Lab

Department of Mental, Physical Health and Preventive Medicine

University of Campania “Luigi Vanvitelli”, via L. Armanni, 5, 80138 – Naples - Italy

Phone +39 0815666008 - giovanni.cirillo@unicampania.it

Issue raised by Reviewer 1

  1. Typographical errors and grammar mistakes must be corrected. 

R. we have thoroughly revised the draft, correcting typo and errors

  1. There are several literature published on inflammatory bowel disease, how you justify this article is different from them?

R: we aim to provide an up-to-date paper on the way central and enteric nervous systems interact and contribute to the immune responses and to the pathogenesis of the inflammatory bowel disease, proposing the basics for the therapeutic application of vagus nerve stimulation.

  1. Authors suggested the targeting therapeutic approaches against IBD. Available drugs have side effects therefore; scientific community looks forward the therapeutic options. Polyphenol regulated gut microbiota approach is a best strategy. Include literature on polyphenol approaches, these components contain immuno-modulatory and antioxidant properties. So, in this review, elaborate the polyphenol effect in therapeutic options against IBD. Add another figure which shows the polyphenol induced gut microbiota modification in vagus nerve stimulation including its molecular mechanism.

R: we thank the reviewer for these comments. In the revised manuscript, we have added a section paragraph and a figure regarding the role of the polyphenols in IBD.

Reviewer 2 Report

The manuscript reviews in detail the biology of the immune system, enteric nervous system and pathogenesis of inflammatory bowel diseases. There is a particular focus on the role and use of vagus nerve stimulation on the immune response in the gut. Although the abstract suggests the intent of the review is to discuss a personalised therapeutic approach for IBD patients using VNS technology, there was little mention of this.

The review was very well written, figures were very high quality and the concept of relating molecular/cellular changes in the gut immune system to VNS is novel and would further advance the field. However, my main concern is the predominance of text that ‘summarises’ information so that the majority of the review reads much like a text book. There is minimal mention of original experiments, species used, mention of studies that contradict the status quo etc. Furthermore, the discussion on VNS was largely irrelevant, with too much focus placed on the effects of VNS in the brain, and little mention of how VNS could be used to provide a personalised therapy to IBD patients. A major re-write is required.  

MAJOR

1.       There is a distinct lack of original evidence discussed throughout the manuscript. For example, lines 185-191 discuss ‘evidence’ of cross talk between enteric neurons and other anatomical structures. This is interesting and key to the review and should be expanded For example, what species showed this? How did they show cross talk-calcium imaging/ephys? What does this mean for VNS? etc

2.       Discuss the increased use of antibiotics as a risk factor of developing IBD

3.       Increasing evidence from De Jonge’s lab shows a role of the sympathetic nervous system in animal models of IBD (Willemze et al., 2018; Willemze et al., 2019). These experiments should be discussed and acknowledged

-          There are a number of key authors you have not included that should be considered in this review:

o   Terry L Powley: World lead in gut neural anatomy

o   H. R Berthoud: World lead in vagal anatomy and digestion

o   Zanos et al., 2018 (PNAS): that vagal afferents can identify between different pro-inflammatory cytokines

o   Joel Bornstein and John Furness: Both world experts in gut physiology and digestive diseases

o   Bruno Bonaz: published the first clinical trial of VNS as a treatment of IBD

-          Figure 2 seems redundant – suggest cutting

-          Figures/figure legends in general lack explanation in the legend. Most figures require a key or additional labels to explain the anatomy/cell types

-          There is little to no description of the vagal nerve innervation anatomy in the gastrointestinal track (6.1). This is critical given you discuss VNS as a therapy.

-          I am unsure why section 7.1 focuses on VNS as a treatment of cancer, heart failure, AD and metabolic syndrome. There is no mention of IBD or rheumatoid arthritis (autoimmune disease). This is irrelevant and text around the human and animal studies supporting use of VNS for IBD treatment should be included instead.

-          As mentioned above, expand and further discuss sympathetic role (lines 430-438)

Author Response

Response to reviewers

Prof. Dr. Naweed I. Syed

MDPI - Cells

Section Editor-in-chief

Manuscript ID: cells-1908669

Dear Editor,

we have greatly appreciated the prompt and constructive revisions that our paper has received. We are now sending a revised version of the manuscript, having addressed the reviewers’ comments.

Therefore, we hope that the responses and the revised manuscript are satisfactory, and meet with your approval.

We look forward to your decision.

Your sincerely,

                                                                                             Giovanni Cirillo

Prof. Giovanni Cirillo, MD, PhD

Division of Human Anatomy, Neuronal Networks Morphology and Complex Systems Biology Lab

Department of Mental, Physical Health and Preventive Medicine

University of Campania “Luigi Vanvitelli”, via L. Armanni, 5, 80138 – Naples - Italy

Phone +39 0815666008 - giovanni.cirillo@unicampania.it

Issue raised by Reviewer 2

  1. The manuscript reviews in detail the biology of the immune system, enteric nervous system and pathogenesis of inflammatory bowel diseases. There is a particular focus on the role and use of vagus nerve stimulation on the immune response in the gut. Although the abstract suggests the intent of the review is to discuss a personalised therapeutic approach for IBD patients using VNS technology, there was little mention of this. The review was very well written, figures were very high quality and the concept of relating molecular/cellular changes in the gut immune system to VNS is novel and would further advance the field. However, my main concern is the predominance of text that ‘summarises’ information so that the majority of the review reads much like a text book. There is minimal mention of original experiments, species used, mention of studies that contradict the status quo etc. Furthermore, the discussion on VNS was largely irrelevant, with too much focus placed on the effects of VNS in the brain, and little mention of how VNS could be used to provide a personalised therapy to IBD patients. A major re-write is required. 

R. we thank the reviewer for these general comments and for appreciating our work. According to these, we have completely revised the draft, critically presenting previous data and experiments.

2. Discuss the increased use of antibiotics as a risk factor of developing IBD

R. we have added a paragraph, according to this suggestion.

  1. Increasing evidence from De Jonge’s lab shows a role of the sympathetic nervous system in animal models of IBD (Willemze et al., 2018; Willemze et al., 2019). These experiments should be discussed and acknowledged

R. we thank the reviewer for this suggestion, we have discussed the role of sympathetic nervous system.

  1. There are a number of key authors you have not included that should be considered in this review:
  • Terry L Powley: World lead in gut neural anatomy
  • H. R Berthoud: World lead in vagal anatomy and digestion
  • Zanos et al., 2018 (PNAS): that vagal afferents can identify between different pro-inflammatory cytokines
  • Joel Bornstein and John Furness: Both world experts in gut physiology and digestive diseases
  • Bruno Bonaz: published the first clinical trial of VNS as a treatment of IBD

R. we are grateful for these suggestions, we have added the missing key references.

  1. Figure 2 seems redundant – suggest cutting; Figures/figure legends in general lack explanation in the legend. Most figures require a key or additional labels to explain the anatomy/cell types

R. we have added one figure, modified the others and add significant details to the figures and to the legends.

  1. There is little to no description of the vagal nerve innervation anatomy in the gastrointestinal track (6.1). This is critical given you discuss VNS as a therapy. Expand and further discuss sympathetic role (lines 430-438)

R. we have added significant details regarding the vagal nerve innervation anatomy in the gastrointestinal track and we have expanded the discussion of the sympathetic role

  1. I am unsure why section 7.1 focuses on VNS as a treatment of cancer, heart failure, AD and metabolic syndrome. There is no mention of IBD or rheumatoid arthritis (autoimmune disease). This is irrelevant and text around the human and animal studies supporting use of VNS for IBD treatment should be included instead.

R. we have rephrased and extended the discussion, according to these comments.

Round 2

Reviewer 1 Report

The article was revised as per suggestion. Now, it can be considered for publication. 

Reviewer 2 Report

Thank you for addressing my comments.